On the Andean genus Leschenius (Coleoptera: Curculionidae: Entiminae): Updated phylogeny, with a new species from Ecuador, discovery of males, and larval description of the potato weevil Leschenius vulcanorum

del Río María Guadalupe gdelrio@fcnym.unlp.edu.ar guadalupedelrio@yahoo.com 1
Marvaldi Adriana E. marvaldi@fcnym.unlp.edu.ar 1
División Entomología, Museo de La Plata, Universidad Nacional de La Plata, CONICET , La Plata , Buenos Aires , Argentina
Morrone Juan J.
Electronic publication date: 2022 Feb 9
Publication date: 2022
Volume: 10
Electronic Location ID: e12913
Received 2021 Nov 26; Accepted 2022 Jan 19
Copyright: ©2022 del Río and Marvaldi
Copyright year: 2022
Copyright holder: del Río and Marvaldi
License: This is an open access article distributed under the terms of the Creative Commons Attribution License, which permits unrestricted use, distribution, reproduction and adaptation in any medium and for any purpose provided that it is properly attributed. For attribution, the original author(s), title, publication source (PeerJ) and either DOI or URL of the article must be cited.
License URL: https://creativecommons.org/licenses/by/4.0/

Keywords: South America, Neotropical region, Andean species, Immature stages, Parthenogenetic weevils, Sp. nov, Leschenius bifurcatus, Leschenius ventrilingulatus

Funding: Agencia Nacional de Promoción Científica y Tecnológica (ANPCYT) PICT 2016-2798 PICT 2019-4229 Universidad Nacional de La Plata (UNLP) # 11/N852 This work was supported by “Agencia Nacional de Promoción Científica y Tecnológica” (ANPCYT) through grants PICT 2016-2798 and PICT 2019-4229, and by “Universidad Nacional de La Plata” (UNLP) through grant # 11/N852. The funders had no role in study design, data collection and analysis, decision to publish, or preparation of the manuscript.

==============================
The weevil genus Leschenius del Río (Curculionidae: Entiminae: Naupactini) is distributed in the northern Andes, in Colombia and Ecuador. Among its species, L. vulcanorum stands out as an important pest of potatoes in its parthenogenetic form, which is known as “tiroteador de la papa”. In this study, the adult male and the larval stage (first and mature larvae) of L. vulcanorun are described and illustrated for the first time. A description of the male of Leschenius bifurcatus is also provided. A new bisexual species was discovered, Leschenius ventrilingulatus sp. nov., and is described from Ecuador. An updated phylogenetic analysis was performed, including the new species, with results indicating a sister group relationship between L. ventrilingulatus and L. vulcanorum. They can be distinguished because the former is usually of smaller size and is covered by denser and thicker setae, has shorter antennae, a subcylindrical shape of the pronotum, shorter elytra (about 1.5×longer than wide at base), the female has ventrite 4 with a posterior rounded projection, and posterior margin of ventrite 5 subacute, not excavated. This paper also includes lectotype designations, a revised key to all known species of Leschenius, habitus photos of males and females, illustrations of genitalia, and a distribution map.

Introduction

The weevils of the South American genus Leschenius del Río (Curculionidae, Entiminae, Naupactini) range in the highlands of Colombia and Ecuador, at approximately 1,800 to 5000 m.a.s.l. (del Río, Marvaldi & Lanteri, 2012). According to the phylogeny of the Naupactini tribe by Lanteri & del Río (2017), the genus Leschenius belongs to the so-called “Andean group”, a clade of genera distributed in the Andes: (Asymmathetes Wibmer & O’Brien (Amphideritus Schoenherr (Leschenius (Amitrus Schoenherr (Obrieniolus del Río (Melanocyphus Jekel, Trichocyphus Heller)))))), supported by the pro-femora about as wide as meta-femora and by some features of the vestiture like the scarcity of scales and the elytral setae being either erect and long or absent. The “Andean group” belongs to a larger clade, defined by the reduction of the elytral humeri and metathoracic wings, that includes most genera often related to Pantomorus Schoenherr sensu lato, like Atrichonotus Buchanan, Aramigus Horn, Phacepholis Horn, Parapantomorus Emden (Lanteri & del Río, 2017).

Leschenius is recognized by the black, denuded, and shiny integument, the well-developed denuded ridge at the apex of the rostrum, bordering the mandibles (pre-epistome), the pronotal base “V” shaped, the elytral base curved backward, the reduction of the metathoracic wings, and by the procoxae separated and situated much closer to the anterior than to the posterior margin of the prosternum. Mixed in the series of L. vulcanorum (Kirsch), we found some specimens which differ in some diagnostic characters such as the length of the elytra, the density of the vestiture, and the shape of the female ventrite 4. After close examination, we concluded that these specimens correspond to a new bisexual species, L. ventrilingulatus del Río & Marvaldisp. nov., close to L. vulcanorum. We also found male specimens, previously unnoticed, as belonging to L. vulcanorum. Finally, and despite its great economic importance as a potato pest, we realized there was not a detailed larval description for this species, or any representative of Leschenius.

The purpose of this contribution was to provide a systematic update of the genus Leschenius, including descriptions of a new species, the larva of L. vulcanorum, the males of L. vulcanorum and L. bifurcatus del Río, Marvaldi & Lanteri, along with lectotype designations, updated phylogenetic analysis and a revised key to all known species of the genus.

Materials & Methods

The study was based upon the examination of adult specimens borrowed from the following institutions: Charles O Brien collection, now housed at Arizona State University (ASUCOB, Tempe, USA), The Natural History Museum (BMNH, London, UK), Museo de La Plata (MLPC, La Plata, Argentina), Muséum National d’Histoire Naturelle, (MNHN, Paris, France), Museum für Tierkunde, (MTD, Dresden, Germany), Museum für Naturkunde (ZMB, Berlin, Germany).

Immature stages. The slide-mounted larval specimens are deposited at the MLPC, labelled with the data of this article. Techniques for dissection of larvae, terminology and abbreviations herein applied correspond to Marvaldi (1998).

Dissections of genitalia were made according to standard entomological techniques, and characters were drawn using a camera lucida adapted to a stereoscopic microscope (Nikon SMZ800). All measurements were taken with an ocular micrometer attached to the same microscope, and their abbreviations used in the description are as follows: L, maximum length; LA, length of antennae; LB, length of body; LE, length of elytra; LP, length of pronotum; W, maximum width; WRa, width of rostrum measured across apex (excluding scrobes); WRb, width of rostrum at anterior margins of eyes. The terminology used for the morphological structures follows Marvaldi et al. (2014), Lanteri & del Río (2017) and the glossary of weevil characters by Lyal (2021). The terminology used for the sculpture follows that of Harris (1979).

The electronic version of this article in Portable Document Format (PDF) will represent a published work according to the International Commission on Zoological Nomenclature (ICZN), and hence the new names contained in the electronic version are effectively published under that code from the electronic edition alone. This published work and the nomenclatural acts it contains have been registered in ZooBank, the online registration system for the ICZN. The ZooBank LSIDs (Life Science Identifiers) can be resolved and the associated information viewed through any standard web browser by appending the LSID to the prefix  http://zoobank.org/. The LSID for this publication is: urn:lsid:zoobank.org:pub:348600A7-0721-4BC9-A3FD-CB5CBDC55954. The online version of this work is archived and available from the following digital repositories: PeerJ, PubMed Central SCIE and CLOCKSS.

Phylogenetic analysis

For the phylogenetic analysis of Leschenius, the list of morphological characters and the data matrix by del Río, Marvaldi & Lanteri (2012) were modified to include the new species as a terminal taxon as well as new information about the males of Leschenius vulcanorum and L. bifurcatus.

For the inclusion of the new species in the present analysis, four characters from the list by del Río, Marvaldi & Lanteri (2012) were redefined (chars. 5, 41, 47, and 48), and a new one for the male genitalia was added (Table 1). The new list consisted of 50 morphological characters of the adults, including 36 from the external morphology and 14 from the female (9) and male terminalia (5). The data matrix herein analyzed includes 12 terminal taxa, corresponding to seven species of Leschenius plus five outgroup taxa (Table 2) closely related to Leschenius according to Lanteri & del Río (2017). All characters were treated as non-additive and analyzed under equal weights.

Table 1 List of characters, character states and coding.

External morphology	
0. Body size (length in dorsal view, from apex of rostrum to apex of elytra): small (less than 8 mm long) (0); medium sized (between 8-10 mm) (1); large (over 10 mm long) (2).	
1. Elytral vestiture: squamose (0); setose (1); scarce or absent (2).	
2. Elytral setae: absent (0); short, suberect (1); long, erect (2).	
3. Rostrum and forehead: smooth (0); punctuate or foveolate (1); foveolate-strigose (2); coarsely strigose (3).	
4. Pronotum: smooth (0); punctuate or foveolate (1); foveolate-granulose (2); tuberculate (3).	
5. Relative length of rostrum, LR/WRa: more than 1 (0); 0.96-1(1) less than 0.95 (2).	
6. Sides of rostrum: slightly convergent towards apex (WRb/WRa less than 1.4) (0); moderately convergent towards apex (WRb/WRa more than 1.4) (1).	
7. Rostral sulcus: reaching forehead (0); exceeding posterior margin of eyes (1).	
8. Size of epistome: narrow (0); moderately wide (1); very wide (2).	
9. Epistome: depressed (0); elevated (1).	
10. Pre-epistome: absent or reduced (0); well developed (1).	
11. Eyes: strongly convex (0); moderately convex (1).	
12. Length of antennal scape: short, not reaching posterior margin of eyes (0); reaching posterior margin of eye (1); slightly exceeding posterior margin of eyes (2).	
13. Ratio between length of funicle segments 2 and 1, La2/La1: more than 1.5 (0); between 1.1 and 1.49 (1); subequal (2).	
14. Ratio between maximum width and length of pronotum, W/L: less than 1.3 (0); more than 1.3 (1).	
15. Shape of pronotum: subcylindrical (0); slightly conical (1).	
16. Sides of pronotum: almost straight to slightly curved (0); moderately curved (1); strongly curved (2).	
17. Pronotal base: straight (0); curved backwards (1); “V” shaped (2); bisinuate (3).	
18. Projection of lateral angles of pronotum of males: absent (0); present (1).	
19. Ratio between maximum length and width of elytra, L/W: more than 1.5 (0); less than 1.5 (1).	
20. Maximum width of elytra: about middle (0); at posterior third (1); at anterior third (2).	
21. Elytral base: bisinuate (0); straight to slightly curved backwards (1); strongly curved backwards (2).	
22. Humeral angles of males: not projected (0); anteriorly projected (1).	
23. Elytral humeri: moderately prominent (0); slightly prominent to absent (1).	
24. Apical projection of elytra: absent (0); present (1).	
25. Elytral apex: entire (0); slightly divided (1); strongly divided or bifid (2).	
26. Elytral intervals, width: slightly wider than striae (1.5-2x) (0); about same width of striae or slightly slender (1).	
27. Elytral intervals, convexity: flat to slightly convex (0); moderately convex (1); strongly convex (2).	
28. Procoxae, separation: contiguous to slightly separate (0); distinctly separate from each other (1).	
29. Procoxae, position: almost contiguous with anterior margin of prosternum (0); about 2x closer to anterior than to posterior margin (1); less than 2x closer to anterior than to posterior margin of prosternum (2).	
30. Row of denticles on inner margin of tibiae: present on three pairs of tibiae (0); present on pro- and mesotibiae (1); present only on protibiae (2); absent on three pairs of tibiae (3).	
31. Corbel of metatibial apex: broad, squamose (0); narrow to moderately broad, squamose (1); narrow, setose or denuded (2); absent (3).	
32. Apical setal comb of metatibiae: longer than dorsal comb (0); about as long as dorsal comb (1); shorter than dorsal comb (2).	
33. Ratio between length of ventrite 2 and ventrites 3+4 (L2/L3+4): subequal (0); between 1.25-1.5 (1); more than 1.5x (2).	
34. Posterior margin of of ventrite 5 in females: rounded (0); blunt (1); escavate (2); slightly pointed (3).	
35. Posterior margin of ventrite 5 in males: rounded (0); bilobated (1); emarginate (2) blunt (3).	
Female terminalia	
36. Plate of female sternite VIII: subrhomboidal, elongate (basal half longer than apical half) (0); subrhomboidal, not elongate (basal and apical half subequal) (1).	
37. Apodeme of female sternite VIII: less than 2.7x longer than plate (0); more than 2.7x longer than plate (1).	
38. Ovipositor: about as long as to longer than ventrites 1–5 (0); 2∖3 to 3∖4 length of ventrites 1–5 (1); about 1∖2 or less the length of ventrites1-5 (2).	
39. Rows of setae along sides of baculi (ovipositor): absent (0); present (1).	
40. Lenght of spermathecal duct: as long as half of the length of ovipositor (=medium-sized) (0); shorter than half of the length of ovipositor (=short) (1).	
41. Spermathecal body: subcylindrical (0); subglobose (1); globose (2).	
42. Duct-lobe (collum) of spermatheca: conical, very short (0); truncate conical, short (1); tubular (2).	
43. Gland-lobe (ramus) of spermatheca: indistinct to slightly developed (0); well-developed (1).	
44. Cornu of spermatheca: short (0); medium length to long (1); very long (2).	
Male genitalia	
45. Ratio between length of penis apodemes and length of penis body (LAp/Lml): apodemes slightly shorter than penis body, (2/3-3/4) (0); about half as long as penis body (1).	
46. Angle between longitudinal axis of penis body and its apodemes: almost flat (0); obtuse to about 90° (1).	
47. Apex of penis, shape in dorsal view: tapering into a long acute projection (0); slightly pointed (1); rounded, with a pointed projection at apex (2); evenly rounded (3).	
48. Endophallic armature: absent, no distinct sclerotized pieces (0); present, with wing-like sclerotized pieces (1) with long sclerotization like flagelum (2).	
49. Apex of penis, curvature in lateral view: not recurved (0); dorsally slightly recurved (1) dorsally strongly recurved (2).	

Table 2 Data matrix of Leschenius plus five outgroups.

		1	2	
	0	1	2	3	4	5	6	7	8	9	0	1	2	3	4	5	6	7	8	9	0	1	2	3	4	5	6	7	8	9	
Melanocyphus lugubris	2	2	0	1	1	0	0	0	1	0	0	1	0	2	0	0	1	3	1	0	1	0	0	0	1	0	1	2	0	2	
Trichocyphus formosus	2	1	2	0	0	0	0	0	1	0	0	1	0	0	0	0	1	0	-	0	0	0	-	0	0	0	0	0	0	2	
Amitrus nitens	2	2	0	1	[1 3]	0	1	0	1	0	0	1	1	1	0	0	0	1	0	0	0	0	1	0	0	0	1	1	0	1	
Amphideritus vilis	1	1	2	3	2	0	1	0	2	1	0	1	1	0	0	0	1	0	0	0	0	0	0	0	0	0	1	0	0	1	
Asymathetes pascoei	0	0	1	2	1	1	1	0	0	0	0	1	1	1	0	0	2	1	0	1	2	0	0	0	0	0	0	1	1	1	
Leschenius rugicollis	1	2	1	3	2	1	0	1	0	0	1	0	1	1	0	1	0	2	1	1	0	2	1	1	1	1	1	0	1	0	
Leschenius nigrans	2	2	1	2	1	1	0	1	0	0	1	1	1	0	0	1	1	2	1	0	2	1	1	1	1	2	0	0	1	0	
Leschenius vulcanorum	0	2	1	2	1	2	1	0	0	0	1	1	2	1	1	1	1	2	0	1	0	1	0	1	0	0	1	0	1	0	
Leschenius manueli	1	2	1	3	2	1	0	1	0	1	1	0	1	2	1	1	1	2	0	1	0	2	1	1	1	0	0	0	1	0	
Leschenius bifurcatus	2	2	1	1	1	1	0	1	0	1	1	1	2	2	0	0	0	0	-	0	0	1	0	1	1	2	1	0	1	0	
Leschenius silviae

Lescheniu ventrilingulatus	1

0	2

[1 2]	1

1	2

2	1

1	1

2	0

1	1

0	0

0	0

0	1

1	1

1	2

2	0

2	1

1	0

0	1

1	2

2	1

0	1

1	2

0	2

1	1

0	1

1	1

0	0

0	1

1	0

0	1

1	0

0	
	3	4	
	0	1	2	3	4	5	6	7	8	9	0	1	2	3	4	5	6	7	8	9	
Melanocyphus lugubris	3	3	1	0	0	–	1	0	1	0	–	2	0	1	0	0	0	1	0	0	
Trichocyphus formosus	0	0	0	0	1	–	1	0	1	1	0	0	0	0	1	-	-	-	-	-	
Amitrus nitens	1	1	1	0	[0 1]	0	1	0	2	1	1	0	1	0	1	0	0	2	1	0	
Amphideritus vilis	2	0	2	0	3	0	1	1	1	0	0	0	1	0	1	0	0	2	0	0	
Asymathetes pascoei	2	1	2	2	1	2	1	1	0	0	0	0	1	0	2	0	0	1	0	0	
Leschenius rugicollis	1	1	1	1	3	1	1	0	1	0	0	0	1	0	2	0	0	3	1	0	
Leschenius nigrans	2	0	0	1	3	3	1	0	1	1	0	0	1	0	2	1	1	1	1	0	
Leschenius vulcanorum	2	1	1	1	2	3	0	0	1	0	0	0	1	0	1	0	0	0	2	1	
Leschenius manueli	2	2	1	1	3	2	1	1	1	1	1	0	1	1	1	0	0	3	2	0	
Leschenius bifurcatus	2	3	1	1	0	1	0	0	1	1	1	0	1	1	2	1	0	1	1	2	
Leschenius silviae	1	2	0	1	2	3	1	0	1	0	0	1	2	0	2	1	1	1	0	0	
Leschenius ventrilingulatus	2	1	1	–	3	1	1	1	1	1	0	0	1	0	1	0	0	0	2	1	

A cladistic analysis was conducted with TNT v1.5 (Goloboff & Catalano, 2016), using the “traditional search” algorithm, with 100 random addition sequences, Tree Bisection and Reconnection (TBR) branch swapping, holding 10 trees during each replication. The most parsimonious tree was rooted with Melanocyphus lugubris. Clade stability was evaluated with 1000 replication Bootstrap (BT) (Felsenstein, 1985), support values over 50% were indicated below branches. The total length (L), the consistency index (CI) (Kluge & Farris, 1969), and the retention index (RI) (Farris, 1989) of the most parsimonious trees (MP tree) were calculated excluding the uninformative characters. The character changes were mapped on the tree using fast (ACCTRAN) optimization with WINCLADA1.00.08 (Nixon, 2002).

Results

Cladistics

The analysis yielded one most parsimonious tree (L = 155 steps, CI = 0.56, RI = 0.53) (Fig. 1). In the cladogram, Asymmathetes pascoei is the sister group of Leschenius, relationship that is supported by several synapomorphies (at least 10 exclusive and one homoplastic character changes, shown in Fig. 1). Leschenius is monophyletic and includes the new species, L. ventrilingulatus, sharing the synapomorphies of the genus: the well-developed pre-epistome (character 10.1); the ‘V’–shaped pronotal base (character 17.2); the slightly prominent to absent elytral humeri (character 23.1); and the procoxae almost contiguous with anterior margin of prosternum (character 29.0). Leschenius is also supported by five homoplastic character states: antennal scape slightly exceeding posterior margin of eyes (character 12.2, with reversal to 12.1 in L. nigrans–L.manueli clade); funicle segments 2 and 1 subequal (character 13.2, with reversal to 13.1 in L. vulcanorum and L. rugicollis); elytral base straight to slightly curved backwards (character 21.1) with evolutionary transition to 21.2 in L. nigrans–L.manueli clade (apomorphic with a reversal to 21.1 in L. nigrans). The latter is an important character for Leschenius because all the other genera of the “Andean group” have the elytral base bisinuate (21.0).

Figure 1 MP tree for the genus Leschenius plus five outgroups members of the Andean Clade sensuLanteri & del Río (2017).

The MPT shows the phylogenetic position of the new species, Leschenius ventrilingulatus. Black circles homology, white circles homoplasy. Numbers below branches are >50% Bootstrap values.

The genus Leschenius has two main clades, named A and B in Fig. 1. Clade A is well supported and includes the new species, L. ventrilingulatus as sister of L. vulcanorum, a relationship supported by the very short rostrum (character 5.2), the relatively wide pronotum (character 14.1), the short cornu of spermatheca (character 44.0); the penis with its apex tapering into a long acute projection (character 47.0), dorsally slightly recurved (character 49.1), with a long flagelum like sclerotization in the endophallus (character 48.2). Clade B includes the remaining five species of the genus, supported by the rostral sulcus exceeding posterior margin of eyes (character 7.1), corbel of metatibial apex narrow, setose or denuded (character 31.2); penis apodemes half as long as penis body (character 45.1), and by some homoplastic characters such as medium sized body (character 0.1), sides of rostrum slightly convergent towards apex (character 6.0), and presence of apical projection of elytra (character 24.1). In clade B, L. bifurcatus is the sister group of the remaining species, which form a clade defined by the elytral base strongly curved backwards (character 21.2) and by the homoplastic characters: antennal scape reaching posterior margin of eye (character 12.1), pronotum slightly conical (character 15.1) with lateral angles projected in males (character 18.1), and humeral angle of males anteriorly projected (character 22.1). They are grouped in two sister subclades, one including L. nigrans and L. silviae, defined by one synapomorphy, the obtuse angle between the longitudinal axis of penis and its apodemes (character 46.1), and four homoplastic character states: funicle segment 2 more than 1.5 times longer than segment 1 (character 13.0), maximum width of elytra at anterior third (character 20.2), apical comb of metatibiae longer than dorsal comb (character 32.0), and blunt posterior margin of ventrite 5 in males (character 35.3). The other subclade includes L. rugicollis and L. manueli and is supported by two synapomorphies, strongly convex eyes (character 11.0), and apex of median lobe evenly rounded (character 47.3), plus three homoplastic character states, the rostrum and forehead coarsely strigose (character 3.3), pronotum foveolate-granulose (character 4.2), and penis apodemes slightly shorter than penis body (character 45.0).

Taxonomy

Leschenius del Río 2012	
Leschenius del Río indel Río, Marvaldi & Lanteri, 2012: 55.	

Most characters of the following larval description, based on specimens of L. vulcanorum, may apply to the genus Leschenius. Terminology as in Marvaldi (1998).

Description of larvae. Mature larva. Body robust, widest at thorax and first abdominal segments. Cuticle asperities present on ventral areas and absent on lateral and dorsal areas. Head (Fig. 2A). Deeply retracted into thorax, longer than wide; posterior margin ogival; posterior half unpigmented, with softer integument and without setae; all setae shifted anteriorly, placed on anterior third. Epicranial line more than 0.5 x the length of head capsule. Frontal lines and endocarina absent. Hypopharyngeal bracon with paramedian maculae. Postoccipital condyles obtuse, hyaline. Setae (Fig. 2A): fs4,5, des5, and les2 long, subequal situated on anterior cephalic fifth; des1, des3 shorter but well developed; fs1,2,3, des4, pes1-4 minute; les1 reduced; vcs1 longer than vcs2, both short. Stemmata absent. Antenna (Fig. 2B) with sensorium about 2.5 x wider than long, truncate at apex. Labrum (Fig. 2C) with lms1,2,3, subequal, lms1 slightly less widely separated than lms2. Epipharynx (Fig. 2D) with mes1 less separated than mes2; epipharyngeal sensilla as single units (not as sensillum clusters), one pair between mes1 and mes2 but closer to mes2, and another pair close to bases of labral rods; labral rods (Fig. 2D) ax shaped, bifurcate, with one arm reaching mes1 and the other mes2. Mandibles (Fig. 2E) with mds1 slightly longer than mds2, both transversely placed within the scrobe; mds2 exterior and slightly basal to mds1. Maxillae (Figs. 2F–2G) with spinules on dorsal surface of external margin of stipes and below mala and palpus; maxillary mala with a row of eight dms and with four vms. Labium (Fig. 2F) with premental sclerite well sclerotized, with posterior extension truncate and expanded at apex, anterior extension slender. Thorax (Fig. 3A). Spiracle (Fig. 3E) ellipsoidal, without airtubes. Pronotum (Fig. 3A) with nine setae. Meso- and metathorax with pds3 distinctly longer than others; alar area with two as. Pedal areas of thoracic segments (Figs. 3A–3B) each with nine setae: seta z conspicuous; setae x and y subequal; u smaller than v; v smaller than w; seta v’ present and rather conspicuous; small x’ distinct; a pair of additional anterosternal microsetae occasionally present in front of each pedal area. Abdomen (Figs. 3A, Figs. 3C–3D): Spiracles (Fig. 3E) elliptical, 2x smaller than thoracic one, without airtubes. Segments AI-VII (Figs. 3A, 3C) with five pds, pds3 and pds5 the longest; spiracular area with only ss2 distinct and progressively placed closer to postdorsum, ss1 vestigial or absent. AVIII with four pds, lacking the homologous pds2 of preceding segments; ss indistinct. Abdominal apex (Fig. 3D) modified, with transverse posterior sclerotized ridges on dorsum, pleura and sternum of AIX; AIX with four ds including a seta ds’, placed lateral to ds1; laterally with two ls strongly unequal, the longest on sclerotized ridge; AX terminal, four-lobed, each lateral anal lobe with three minute setae, the outermost larger.

Figure 2 Leschenius vulcanorum, larvae. Head morphology.

(A–G) Mature larva. (A) Head, dorsal. (B) Left antenna. (C) Clypeus and labrum. (D) Epipharynx. (E) Mandible, dorsal. (F) Maxilla and labium, ventral. (G) Maxilla, dorsal. (H–J) First instar larva. (H) Head, dorsal. (I) Head, partial, ventral. (J) Mandible, dorsal. Scales A, C–G = 0.5 mm; B, H–J = 0.1 mm.

Figure 3 Leschenius vulcanorum, larvae. Thoracic and abdominal morphology.

(A–E) Body parts and chaetotaxy, mature larva. (A) Prothorax, mesothorax, metathorax and abdominal segment I, one side from mid-dorsum to mid-ventral. (B) Detail of pedal area. (C) Abdominal segment IV, dorso-lateral parts. (D) Abdominal apex, segments IX and X, caudal view. (E) Spiracles from thorax and abdominal segments I, IV and VIII. (F) Spiracles from thorax and abdominal segments I, IV and VIII, first instar larva. Scales A–D = 1 mm, E=0.5 mm; F=0.1 mm.

First instar larva (Figs. 2H–2J, 3F). Head (Figs. 2H–2I) only slightly retracted into thorax, slightly longer than wide; major cephalic setae placed on anterior half, des2 and les1 less reduced than in older larvae; des1 minute (smaller than in mature larvae). Frontal lines weakly distinct. Anterior and posterior stemmata distinct, as dark pigmented spots. Antennal sensorium prominent and projected outwards. Mandibles (Fig. 2J) with mds1,2 strongly unequal, mds1 about 5x longer than mds2. Thorax. Spiracle (Fig. 3F) bicameral with annulated airtubes; pedal area with setae z, and v’ relatively small, seta w relatively very long and spatulate or blunt at apex. Abdomen. Spiracles (Fig. 3F) bicameral, with airtubes having a smaller number of rings than thoracic one; abdominal apex not distinctly sclerotized.

Remarks. The characters mentioned above for the first larva, newly hatched, are exclusive of the first instar (see also Marvaldi & Loiácono, 1994). Additional differences between early and older instar larvae involve relative dimension of structures, like the antennal sensorium, which are relatively much larger in the first instar; the pigmentation and level of sclerotization of body areas tend to increase in successive instars; there are larger differences in length between setae of different body areas in early instars.

Comparative notes. The larval characters given herein for the genus Leschenius are in full agreement with those defining the tribe Naupactini (Marvaldi & Loiácono, 1994) or Naupactus and allied genera (Marvaldi, 1998). Two apparently distinct features of the mature larva studied of L. vulcanorum are the head capsule with des1 well developed (in known mature larvae of other Naupactini des1 is minute or very small); also, setae x and y of pedal areas of thoracic segments are subequal (in other Naupactini as Naupactus, seta x is distinctly smaller than y). Larval characters deemed as diagnostic for the species are given below.

Notes on type material. Four paratypes of the species of Leschenius described in del Río, Marvaldi & Lanteri (2012) were finally deposited in the MLPC instead of the collection mentioned in the original publication: one paratype of L. bifurcatus del Río, Marvaldi & Lanteri, with labels ‘ECUADOR, PICHINCHA, POMASQUI, RUNICUCHO, 2400 m, 6–XII–1993, K. Volbracht’; two paratypes of L. manueli del Río, Marvaldi & Lanteri, with labels ECUADOR, AZUAY, VIA CUENCA LOJA, 5 km DE ONA, 13–I–1997, A. Paucar’; and one paratype of L. silviae del Río, Marvaldi & Lanteri, with label ‘Cuenca Jesta’.

Leschenius vulcanorum (Kirsch, 1889)	
(Figs. 2, 3 and 4, Figs. 5A, 6G, 6K, Fig. 7)	
Canephorus vulcanorum Kirsch 1889: 17; Strand 1943: 96 (Canephorulana);	
Kuschel 1955: 277 (Amitrus); Kuschel in Wibmer & O’Brien 1986: 53 (Asymmathetes) (Fig. 4A).	
Amphideritus brevis Oliff 1891: 68; DallaTorre, Emden & Emden 1936: 14 (Macrostylus); Kuschel 1955: 277 (Amitrus) (syn. of A. vulcanorum); Kuschel in Wibmer & O’Brien 1986: 53 (Asymmathetes); del Río, Marvaldi & Lanteri, 2012: 60 (Leschenius) (Fig. 4B).	
Amphideritus pigmaeus Oliff 1891: 68; Dalla Torre, Emden & Emden 1936: 14 (Macrostylus); Kuschel 1955: 277 (Amitrus) (syn. of A. vulcanorum); Kuschel in Wibmer & O’Brien 1986: 53 (Asymmathetes) (Fig. 4C).	
Caulostrophus aequatorialis Kirsch 1889: 13; Dalla Torre, Emden & Emden 1939: 319 (Macrostylus [Amphideritus]); Kuschel in Wibmer & O’Brien 1986: 53 (Asymmathetes) (Fig. 4D), syn. n.	

Figure 4 Types, females, corresponding to the four species names of Leschenius vulcanorum.

(A) Lectotype of Canephorus vulcanorum Kirsch 1889, MTD. (B) Lectotype of Amphideritus brevis Oliff 1891, MNHN. (C) Lectotype of Amphideritus pigmaeus Oliff 1891, MNHN. (D) Lectotype of Caulostrophus aequatorialis Kirsch 1889, MTD. Scales 1 mm.

Figure 5 Photographs of Leschenius species.

(A) Leschenius vulcanorum, male, habitus dorsal view. (B-F) Leschenius ventrilingulatus sp. nov. (B) holotype female, habitus dorsal view. (C) Paratype male, habitus dorsal view. (D) Holotype female, ventral view. (E) Holotype female, habitus lateral view. (F) Holotype female, head, frontal view. (G-H) Leschenius bifurcatus. (G) Female, habitus dorsal view. (H) Male, habitus dorsal view. Scales 1 mm.

Figure 6 Morphological characters, and female terminalia of Leschenius ventrilingulatus, and male terminalia of Leschenius species.

(A–F) Leschenius ventrilingulatus. (A) Female, left antenna. (B) Female, sternite VIII. (C) Detail of plate of sternite VIII. (D) Female genitalia, ventral view. (E) Detail of distal third of ovipositor. (F) Spermathecae with spermathecal duct. (G-I) Aedeagus, lateral view. (J) Detail of apex, lateral view, left: phenotype from Imbabura, right: typical phenotype. (K-M) terminal portion of tube, ventral view. (G, K) L. vulcanorum. (H, L) L. ventrilingulatus. (I, J, M) L. bifurcatus. Scales 1 mm.

Figure 7 Distribution map of the seven species of Leschenius.

Ecuador and southern Colombia are shown in detail. Species references: L. nigrans, white circle; L. rugicollis white square; L. vulcanorum, parthenogenetic form white ellipse, bisexual form grey ellipse; L. manueli white triangle; L. bifurcatus black star; L. silviae black triangle; L. ventrilingulatus white star.

Diagnosis and description of female in del Río, Marvaldi & Lanteri (2012). Description of male (Fig. 5A). Smaller than female (4.0–6.3 mm; females 5.3–8.7 mm); rostrum shorter (L/Wa: 0.76–0.84); less convergent towards apex (Wb/Wa, 1.15–1.30); antennal club more elongate (L/W, 2.5–2.8); pronotum (W/L: 1.25–1.35), wider than the elytra and longer than in females, with sides more arcuate; elytra slightly shorter (L/W, 1.32–1.40) with apex not divided, more rounded; metatibiae with larger mucro than in females; posterior margin of ventrite 5 blunt. Genitalia (Figs. 6G, 6K). Median lobe slightly curved in lateral view, tapering towards apex, with subacute, dorsad slightly recurved hook-like apex; penis as long as abdomen; apodemes slightly shorter than median lobe (0.7×); endophallus armed with minute spicules and with a slightly sclerotized flagellum.

Larval stage. One mature larva and four submature larvae, as well as associated adults of L. vulcanorum, were collected from the following locality: Colombia, Municipio de Sibaté, vereda el Romeral, 4°26′3″N, 74°14′8″O (3100 masl), J.E.C. Gomez leg., 2009. Additionally, 10 first instar larvae were obtained from eggs deposited by some of the collected adults kept in captivity. After comparison with larvae known for other species in tribe Naupactini (Marvaldi & Loiácono, 1994; Marvaldi, 1998) the following combination of characters can be suggested as diagnostic for the species L. vulcanorum.

Mature larva (Figs. 2G, 3A–3E). Maximum head width 2.2 mm. Setae fine, brown. Head yellowish, intense yellow on anterior margin of frons, about 1.3× longer than wide. Cephalic setae (Fig. 2A): des1 well developed (although shorter than des3 and those placed on anterior third, fs4, fs5, des5, and les2). Ephipharynx with spinules anteriad and posteriad to the labral rods; epipharyngeal sensilla not in clusters but apparently fused into single units. Pronotum pigmented with pattern of brownish maculae (Fig. 3A). Abdominal apex (AIX) with transverse sclerotized ridges in dorsum, pleura and sternum (Fig. 3D).

Larva 1 (Figs. 2H–2J, 3F). Maximum head width 0.2 mm. Head (Fig. 2A) with des1 minute (like des 4), cephalic setae well developed are des2, des3, des5, fs4, fs5, les1, and les2; des1 slightly more widely separated than fs4. Clypeus with setae subequal. Labrum with lms2 somewhat more widely separated than lms1.

Type material examined. Lectotype of Canephorus vulcanorum Kirsch, female, Ecuador Tunguragua, 3800, Canephorus vulcanorum typus Kirsch, MTD, (Fig. 4A), here designated. Paralectotype of Canephorus vulcanorum, Ecuador, Sangay, 3500m, cotypus, MTD. Lectotype of Amphideritus brevis Oliff, female, Ecuador feet, Ed. Whymper, MNHN, here designated (Fig. 4B). Lectotype of Amphideritus pigmaeus Oliff, female, Chimborazo, Ecuador, 12-13000 ft, Ed Whymper, MNHN, here designated (Fig. 4C). Lectotype of Caulostrophus aequatorialis Kirsch, female, Cotopaxi, 5688, Typus, MTD, here designated (Fig. 4D).

Other material examined. COLOMBIA. No loc., int. Miami, 5-2-85, with cut flowers of Dianthus sp. (1f USNM). ECUADOR. No loc., 11-11-93 (2f USNM). Chimborazo: Chimborazo, S side of Mt, elev 11600 nr Snowline, 19 june 1975 (1f USNM); Colta, 3-VI-05, Ohaus, 9-VII-05 (27f ZMB); Interandin-Hochland, Colta 3500–4000 m, 8-10-VII-1905 Ohaus (1f ZMB), 2,8-VII-1905 Ohaus (2f ZMB); Faldas del Chimborazo, jan-1983, in pine leaves Pinus radiata, adults feed, Lopez col (1f USNM); Guamote, 3-7-1969, en cocoons of alfalfa plants (1f USNM); Guaslam prov, 1-22-60-on bucts of young corn, Merino (2f USNM); Quimiag on maize, Jan 1979 Kirckhy (2f BMNH); Riobamba, 3-VII-1905, unter steinen, Ohaus (17f ZMB), 20-XI-05 (1f ZMB), 27-XI-05 (2f ZMB); Riobamba, Ause de Cubillin, 3500, 5-Vii-05 Ohaus (42f ZMB). Cotopaxi: 71 km W Latacunga under stones May 1, 1978, O’Brien & Marshall (1f 1m MLP; 45 km W Latacunga, under stones May1 1978, O’Brien & Marshall (8f MLPC); 21 km S Latacunga, April 25 1978, CW&L O’Brien & Marshall (1f MLP); 6 km W Latacunga, under stones, May1 1978, O’Brien & Marshall (3f MLP); Latacunga, XI 1981 Onore Brit Mus 1990-214 (1f BMNH); 15 km W entrance PN Cotopaxi, April 30 1978, O’Brien & Marshall (4f MLP); 14 km W entrance PN Cotopaxi, April 30 1978, O’Brien & Marshall (1f MLP); Cotopaxi, P. 13 km S Latacunga along PanAma, XI-3-77, G Noonan, M. Moffett, under clumps soil and grass, rocks, debris-in green grassy field w green short grass ca 2600 m. (13f 1m MLP); Tilipulo, V-III-1981, G. Onore Brit Mus 1985-254 (4f BMNH). Bolívar: Guaranda, X-I-1955, on new corn (7f USNM). Loja. Loja, Ohaus (2f ZMB). Pichincha: 38.8 km NE Quito on PanAm XI-8-77, G Noonan, M. Moffett, under rocks on dirt clumps, in areas with sparse to very sparce short grass ca 2200 m. (1f MLP). Tungurahua: SE end Ambato, XI-1-77, G Noonan, M. Moffett, under rock in fields w short sparce grass, soil dry under stones, ca 2500 m. (2f MLP); 13 km NE Baños, April 26 1978, O’Brien & Marshall (1f MLP); Baños, 1800 m, 9-V-37 Brundage (2f USNM); Baños, X-4-44, EJ Hambleton (1f USNM); Pomasqui, E = 0, Merino, orange trees (4f USNM); Totoras, 7 km SE Ambato, April 26 1978, O’Brien & Marshall (3f MLP). Plus, the material listed in del Río, Marvaldi & Lanteri (2012).

Remarks. In the revision of the genus Leschenius (del Río, Marvaldi & Lanteri, 2012), the type material of the species Asymmathetes aequatorialis (Kirsch) was not seen and we mentioned that this species may also belong to the genus Leschenius. Herein, based on the observation of the type material of all the names related to Leschenius vulcanorum (Figs. 4A–4D), including Caulostrophus aequatorialis Kirsch (Fig. 4D), we establish the synonymy of this name with L. vulcanorum (Fig. 4A). This species is only known from the type material and corresponds to a phenotype within the great variation observed in L. vulcanorum (see Figs. 4A–4D).

Bisexual populations of L. vulcanorum have been so far only seen in Ecuador, near Latacunga locality (Cotopaxi province), and in Ambato locality (Tungurahua province) (Fig. 7). Differences noted between the bisexual and the parthenogenetic populations are related with the body size and morphometrics of the elytra. The bisexual form is usually smaller with slightly shorter elytra.

Host plants. Leschenius vulcanorum was found in association with cabbage Brassica oleracea L. (Brassicaceae), alfalfa Medicago sativa L. (Fabaceae), young corn Zea mays L. (Poaceae), pine Pinus radiata D. Don, orange trees, and with cut flowers of carnation Dianthus sp. It is considered an important pest of potato Solanum tuberosum L. (Solanaceae) in Colombia, and is known as ‘potato shooter’ (del Río, Marvaldi & Lanteri, 2012).

Leschenius ventrilingulatusdel Río & Marvaldi, sp. nov.	
urn:lsid:zoobank.org:act:9A70B8AE-74BF-4631-98A7-A1C814113833	
(Figs. 5B–5F; 6A–6F, 6H, 6L, 7)	

Diagnosis. Leschenius ventrilingulatus is easily distinguished from the remaining species of Leschenius (except L. vulcanorum) by possessing a shorter and less conical rostrum with sides not thickened and elevated, and apex not projected. It is very similar to L. vulcanorum, but distinguished by its size usually small, vestiture of denser and thicker setae (mainly on head, legs and elytra), shorter antennae (with funicular segments 1 and 2 subequal); shape of the pronotum, subcylindrical with anterior margin as wide as posterior margin; elytra shorter (about 1.5×longer than wide at base; 2×in L. vulcanorum); the metatibial apex with narrow corbel; female with ventrite 4 with a posterior rounded projection; and posterior margin of ventrite 5 subacute, not excavated. Female genitalia with plate of sternite VIII sub-rhomboidal, not elongate, with longer apodeme.

Description. Female (Figs. 5B, 5D–5F). Species medium-sized (LB, 5.0–6.0 mm). Tegument visible, dark brown to reddish brown, shiny. Vestiture composed of disperse, pale ocher to cream colored setae, moderately dense, cream-colored decumbent setae-like scales, absent on middle of pronotum (forming wide lateral stripes) on pronotum, grouped on patches on elytra, in some cases devoid of scales along middle line; also present on venter and legs (more abundant on distal third of femora). Rostrum very short (Fig. 5F) (L/Wa, 0.73–0.83), sides moderately convergent towards apex (Wb/Wa, 1.31–1.43), dorsum moderately convex. Forehead foveolate–strigose, with longitudinal striae. Vertex sparsely punctate. Antennae (Fig. 6A) of medium length (LB/LA, 2.50–2.85); scape reaching to slightly exceeding posterior margin of eyes. Funicle with segment 2 about as long as segment 1, both elongated; funicle segments 3–6 slightly longer than wide, and funicle segment 7 as long as wide; club slightly fusiform (L/W, 2.3–2.4). Pronotum (Fig. 5B) slightly subcylindrical, moderately wider than long (W/L, 1.30–1.35), with anterior margin as wide as posterior margin; median groove absent. Scutellar shield subtriangular, large and wide (surrounded by elevated edges), denuded. Elytra (Figs. 5B, 5E) short (L/W, 1.23–1.33), with maximum width on anterior third, slightly convex; base slightly curved backwards on middle; intervals about twice as long as striae, flat; striae with medium-sized punctures, 9–10 slightly closer along posterior two-thirds; elytral apex acute not projected or bifurcate, entire. Legs. Procoxae much closer to anterior than posterior margin of prosternum; protibiae with medium-sized mucro, and row of acute small denticles (six or seven, on distal two-thirds of tibiae); meso- and metatibiae with small mucro and without denticles; metatibial apex with narrow corbel covered with disperse small elongate cream scales; apical and dorsal combs subequal. Abdomen (Fig. 5D). Intercoxal portion of ventrite 1 slightly broader than metacoxal cavities (1.10–1.15×); ventrite 2 longer than ventrites 3 and 4 combined (1.60×without projection; 1.10 along midline); ventrite 4 with a posterior rounded projection; posterior margin of ventrite 5 subacute, not excavated; tergites I–VII membranose. Terminalia. Sternite VIII (Figs. 6B–6C) with plate sub-rhomboidal, not elongate, with tuft of medium-sized and coarse setae, and with shorter setae on apical third; ‘V’-shaped sclerotization with lateral arms reaching two-thirds of plate, and lateral margins sclerotized; apodeme 2.8–3.0×longer than plate. Ovipositor (Figs. 6D–6E) as long as ventrites1–5; with scattered fine short setae on sides of baculi on anterior third; ventral baculi subparallel; styli wide. Spermathecal body (Fig. 6F) sub-cylindrical; collum (duct-lobe) conical, short; ramus (gland-lobe) indistinct; cornu long; spermathecal duct (Fig. 6D) short, half as long as ovipositor, or 3×the maximum width of spermatheca, membranous, moderately wide.

Male (Fig. 5C). Same size as female (4.8–5.5 mm); rostrum less conical (Wb/Wa, 1.32–1.37); antennal club more elongate (L/W, 2.6–2.7); elytra slightly more elongate (L/W, 1.30–1.35); metatibia with larger mucro than in female; ventrite 4 without posterior projection; posterior margin of ventrite 5 blunt. Genitalia (Figs. 6H, 6L). Penis (median lobe) slightly curved in lateral view, tapering towards apex, with dorsad recurved hook-like apex; penis as long as abdomen; penis apodemes slightly shorter than penis body (0.8×); endophallus armed with minute spicules and with a sclerotized flagellum, with a denticulated blade (Fig. 6H).

Morphometrics. Holotype, female: rostrum L/Wa: 0.77, Wb/Wa: 1.31; antenna LB/LA: 2.85, antennal club L/W: 2.31; pronotum W/L: 1.34; elytra L/W: 1.33; LE/LP: 2.6.

Etymology. The specific name refers to the tongue-like projection of the female venter.

Material examined. Holotype. Female, five mm long, with labels as follows “ECUADOR, Totoras,/ 7km SE. Ambato/Apr. 26, 1978 C&L/ O’Brien & Marshall” (MLPC). Paratypes. Same data as holotype (1m ASUCOB); ECUADOR, 6 km W/ Latacunga under/ stones May 1, 1978/ O’Brien & Marshall (1f, 1m MLPC, dissected with genitalia in vial with glicerine); ECUADOR/ Latacunga/ IX- 1981 Onore/ Brit. Mus./ 1990-214 (3f 2m BMNH); ECUADOR, 5km SE./ Pelileo, April/ 26, 1978 CW&LB/ O’Brien & Marshall (1m ASUCOB); ECUADOR/ Catamayo B./ Loja 5.9.05/ F. Ohaus (1f ZMB).

Geographic distribution (Fig. 7). Leschenius ventrilingulatus is endemic of Ecuador, known for Cotopaxi, Loja, and Tungurahua provinces. It is distributed on the interandean region on river basins, between 1250 and 2750 m above sea level (unlike L. vulcanorum which is distributed in higher altitudes 2600–5000 m. a. s. l.). It is sympatric with L. vulcanorum (Fig. 7) in Latacunga (Cotopaxi province) and Totoras (Tungurahua province) at 2600–2750 m. a. s. l.

Remarks. Leschenius ventrilingulatus is the sister species of L. vulcanorum, according to results of the cladistic analysis herein undertaken (Fig. 1).

Leschenius bifurcatus(del Río, Marvaldi & Lanteri, 2012)	
(Figs. 5G–5H, 6I, 6J, 6M, 7)	

Diagnosis and description of female in del Río, Marvaldi & Lanteri (2012). Description of male (Fig. 5H). Similar size as female, slightly smaller (Fig. 5G) (10–13 mm); rostrum (L/Wa, 1.0–1.1; Wb/Wa, 1.30–1.45); antennal club slightly more elongate (L/W, 3.1–3.4); pronotum (W/L, 1.17–1.19); elytra (L/W, 1.60–1.65) with projected apex but not bifurcated, only slightly divided; ventrites 3-5 not bulged as female, posterior margin of ventrite 5 blunt. Genitalia (Figs. 6I, 6J, 6M). Penis body slightly curved in lateral view, tapering towards apex (Fig. 6M), with dorsad strongly recurved hook-like apex (Fig. 6J); penis longer than abdomen (1.25–1.30); penis apodemes much shorter than median lobe (0.4×); endophallus armed with minute spicules and with two long wing-shaped sclerites (Fig. 6I).

Material examined. ECUADOR. No loc, intercept. Port Miami, 16-VI-2004 (1f USNM). Imbabura: 3.9 km N. Ibarra on Pan Am, XI-II-77, Moffet collr, under rocks by rd. in dry area, with sparce grass ca. 2300 (1f, 3m ASUCOB; 1f, 2m MLPC;); Urcuqui, 14-III-62, Merino & Vasquez, in soil nr cotton plants (2m USNM); El Chotar, Mr. Juncal, 1 june-1961, Merino & Vasquez, reared from larvae doing damage to roots of bean plants (1m USNM). Pichincha: Conocoto, 31-Jan-1992, Alvaro Barragan (1m MLPC); Pomasqui, Runicucho, 2400 m., 6 Dec-1993, E. Volbracht (1m PUCE). Plus the material listed in del Río, Marvaldi & Lanteri (2012).

Remarks. The specimens of the population from Ibarra (Imbabura) are slightly different from the type, the female (Fig. 5G) has a wider and shorter pronotum and the elytral apex only slightly bifurcated but strongly projected posteriad; the males have also a wider and shorter pronotum and the penis with a shorter ostium area and a more recurved apex (Fig. 6J).

Host plants. The larvae of L. bifurcatus were found in association with bean plants, Phaseolus vulgaris L. (Fabaceae).

Key to species of Leschenius

Modified from del Río, Marvaldi & Lanteri (2012).

1. Size 4–8 mm (usually 5–7 mm, exceptionally more than eight mm but never more than nine mm); rostrum very short (L/W less than 0.95, usually near 0.8); pronotum with setae forming two feeble lateral stripes; elytral apex not projected……………….……2	
1′.Size 8–12.5 mm; rostrum moderately short (L/W more than 0.96, usually 1); pronotum without or with scattered setae, never forming lateral stripes; elytral apex projected backwards.…………..…………………………………………..…………………….3	

2.Elytra with cream-colored decumbent setae like scales grouped on patches on entire elytra; pronotum subcylindrical, with anterior margin as wide as posterior margin; metatibial apex with narrow corbel; female with ventrite 4 with a posterior rounded projection (Fig. 5D)…. …………………………………………L.ventrilingulatus,sp. nov. (Figs. 5B–5F)	
2′. Elytra devoid of cream-colored decumbent setae like scales, or if present, limited to margins (intervals 9-10); pronotum slightly subconical, with posterior margin wider than anterior margin; metatibial apex with moderately broad corbel (width: 1/3 of the length of the tibial apex); female with ventrite 4 without posterior projection……………… ………L.vulcanorum (Figs. 4A–4D, 5A, see fig. 1 in del Río, Marvaldi & Lanteri, 2012)	
3.Pronotum with posterior margin as wide as anterior margin; elongate elytra (3x or more the length of pronotum); elytral apex, in females strongly projected backwards and distinctly bifid or divided; in males, moderately projected and rounded to slightly divided............................. ……………………………………………………….... ………………L.bifurcatus (see figs. 2–fig. 3 of del Río, Marvaldi & Lanteri, 2012)	
3′.Pronotum with posterior margin distinctly wider than anterior margin; oval elytra (less than 2.8x the length of pronotum, usually between 2.2–2.7x); elytral apex of both males and females moderately projected, entire to slightly divided…………………4	
4. Corbel plate of metatibia broad (width: almost half the diameter of apex of tibiae) ............... ........………L.nigrans (see figs. 9–10 of del Río, Marvaldi & Lanteri, 2012)	
4′. Corbel plate of metatibia narrow (less than quarter apex of tibiae)……………..5	
5. Body length 8.4–9.3 mm, moderately sized; pronotum as wide as elytra in males; elytra 2.2-2.45x the length of pronotum; spermatheca with collum not constricted....6	
5′.Body length 7.8–8.4 mm, small; pronotum distinctly narrower than elytra in both sexes; elytra 2.5–2.75x the length of pronotum; spermatheca with tubular collum, constricted near spermathecal body……………………………………………… ………………L.silviae (see figs. 11–12, fig. 42 of del Río, Marvaldi & Lanteri, 2012)	
6. Pronotum slightly wider than long (W/L: 1.10–1.25); ovipositor without setae on sides of baculi; apex of penis slightly pointed............................................................. ………………….L.rugicollis (see figs. 4–5 of del Río, Marvaldi & Lanteri, 2012)	
6′.Pronotum of males strongly wider than long (W/L: 1.30–1.35); ovipositor with conspicuous setae on sides of baculi; apex of penis rounded....................................... ................................... L.manueli (see figs. 6–8 of del Río, Marvaldi & Lanteri, 2012)	

Discussion

The cladistic analysis led us to uncover evidence for the monophyly of the genus Leschenius and for the relationships among its species. It is worth to note that Asymmathetes pascoei is supported as the sister group of Leschenius, in agreement with del Río, Marvaldi & Lanteri (2012), but differing from results of the analysis by Lanteri & del Río (2017): (Asymmathetes (Amphideritus (Leschenius (Amitrus (Obrieniolus (Melanocyphus, Trichocyphus)))))). However, this is not unexpected as these works had different objectives and thus differ in the taxon and character sampling (e.g. in the analysis of Lanteri & del Río there is only one species representing the genus Leschenius and, on the other hand, the present study was not designed to test intergeneric relationships). Further research on the systematics and phylogeny of the “Andean Group” will require expanding the outgroup and ingroup taxon sampling (e.g., including Obrienolus and more species representative of each genera) and using additional sources of evidence like molecular data.

The description of a new species and more complete information about male characters, allowed us to propose an updated phylogenetic hypothesis of Leschenius, which differs from the previous study by del Río, Marvaldi & Lanteri (2012) regarding the position of L. bifurcatus and relationships within clade B (Fig. 1). In the new phylogeny L. bifurcatus is the sister species of a subclade that contains L. nigrans sister of L. silviae and L. rugicollis sister of L. manueli. Also, the present study led to the discovery of the sister species of L. vulcanorum and the pair L. ventrilingulatus –L. vulcanorum is proposed as the sister group of all remaining species of Leschenius.

Leschenius is distributed in the northern Andes of Ecuador and northern Colombia, approximately 1800–5000 m.a.s.l (Fig. 7), corresponding to the Páramo province of the South American Transition Zone (Morrone, 2014). All species included have a narrow distribution range in Ecuador, except L. vulcanorum that is widely distributed from central to northern Ecuador and extend to southern Colombia. The latter species, L. vulcanorum, along with L. ventrilingulatus, L. bifurcatus, and L. nigrans inhabit in northern Ecuador, and they have been found in sympatry (Fig. 7). The other three species, L. silviae, L. manueli, and L. rugicollis are distributed in the southern provinces of Ecuador. The new species L. ventrilingulatus is distributed on the inter-Andean region on river basins (Fig. 7) and is sympatric with L. vulcanorum at 2500–2750 m.a.s.l., that corresponds to the lowest altitude for L. vulcanorum.

Of particular interest is the occurrence of parthenogenesis within the genus Leschenius, and this mode of reproduction was suggested by del Río, Marvaldi & Lanteri (2012) for the species L. vulcanorum and L. bifurcatus, based on the apparent absence of males in their populations. The examination of additional material in the present study provided new evidence that suggest L. bifurcatus is not parthenogenetic, leaving Leschenius vulcanorum as the only parthenogenetic species. The parthenogenesis of L. vulcanorum was confirmed by laboratory rearing of unmated females that produced viable offspring (del Río, Marvaldi & Lanteri, 2012). Nonetheless, in the present study, we discovered males of L. vulcanorum, indicating that this species also has bisexual populations, so far only seen in three localities in Cotopaxi and Tungurahua provinces of Ecuador. The existence of both sexual and parthenogenetic populations within the species supports the idea of a special kind of parthenogenesis, called “geographical parthenogenesis” (Lanteri & Normark, 1995). In L. vulcanorum, as in other species having this kind of reproduction, the sexual and parthenogenetic forms have different distribution ranges, being the parthenogenetic one more widespread than the respective sexual forms (Vandel, 1931). So far, exemplars of both sexes of this species were collected in just three localities. Morphological differences between the bisexual and the parthenogenetic form are noticed in body size and morphometrics of the elytra, being the bisexual form usually smaller and with slightly shorter elytra. It is important to remark that within the parthenogenetic form the variation in body size and morphometrics is extremely wide (Figs. 4A–4D). Same as other parthenogenetic weevils, including the around 30 parthenogenetic species of the tribe Naupactini (Lanteri & Normark, 1995), in L. vulcanorum the parthenogenesis is also associated with the wingless condition and xeric habitats.

An interesting feature observed in females of the new species described, L. ventrilingulatus (sexually dimorphic) is the ventrite 4 with its posterior margin produced medially into a tapered lamina or tongue-like projection. This characteristic seems to be unique within the tribe Naupactini, and so far, also unknown among Neotropical Entiminae. Although unusual, a modified ventrite 4 is also present in genera of Entiminae inhabiting montane areas of other regions of the word (see Brown, 2017 and references therein): New Zealand [Austromonticola Brown, Chalepistes Brown, and Nicaeana Pascoe], Solomon Islands [Platyacus Faust, (Celeuthetini)], Mauritian Islands [Syzygops Schönherr, 1826 (Ottistirini)]; Chinese Himalayas [Trichalophus caudiculatus (Fairmaire,) (Tropiphorini)], Kashmir and Himalayas [Leptomias Faust, (Tanymecini)], and Central America [Sciomias Sharp (Sciaphilini)]. These structures are hypothesized to have evolved in response to oviposition needs in and beside cushion plants or selected to mitigate the female costs of prolonged mating (Brown, 2017). Although the function has not been studied yet, the first suggestion that these ventral structures may assist in the preparation of oviposition sites in close-packed vegetational structures seems highly plausible: the cushion growth form is a common feature of the alpine vegetation worldwide, where the weevil species with this trait are found.

Concerning the immature stages, the study of the larvae of Leschenius lead us to confirm that the tribe Naupactini is very homogeneous in larval morphology, as suggested by Marvaldi (1998). Also, there seem to be no particular features that could be suggestive as adaptive traits to arid environments. This is not unexpected, since Leschenius larvae, like those of most entimines, are subterranean and then “preadapted” to live in harsh environments.

Finally, additional research with molecular tools will be very important to find out the genetic divergence and evolution of the group, and the role of the parthenogenesis. Moreover, the study of the biology and behavior of L. ventrilingulatus will offer insights into the function of the abdominal structures of the female.

We thank all the specialists and curators that loaned us specimens for study and/or facilitated the examination of the material in their collections.

Additional Information and Declarations

Competing Interests

Author Contributions

Data Availability

New Species Registration

The authors declare there are no competing interests.

María Guadalupe del Rio and Adriana E. Marvaldi analyzed the data, prepared figures and/or tables, authored or reviewed drafts of the paper, and approved the final draft.

The following information was supplied regarding data availability:

The body measurements are available in the article. The material, labelled with all the data, is available in the MLP collection and can be accessed by contacting the curators.

The Curator of the Section Coleoptera of the Entomology Collection of MLP: Dr. María Guadalupe del Río, gdelrio@fcnym.unlp.edu.ar.

The Head Curator of the Entomology Collection of MLP: Dr. Pablo Dellapé, pdellape@ fcnym.unlp.edu.ar.

The following information was supplied regarding the registration of a newly described species:

Publication LSID: urn:lsid:zoobank.org:pub:348600A7-0721-4BC9-A3FD-CB5CBDC55954

Leschenius ventrilingulatus LSID: urn:lsid:zoobank.org:act:9A70B8AE-74BF-4631-98A7-A1C814113833.

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
