# Peer review of "On the Andean genus Leschenius (Coleoptera: Curculionidae: Entiminae): Updated phylogeny, with a new species from Ecuador, discovery of males, and larval description of the potato weevil Leschenius vulcanorum"

_PeerJ, doi:10.7717/peerj.12913_

## Round 0.1 · original submission · Minor Revisions

Please make the corrections suggested by both reviewers and send a revised version of your manuscript.

·

Basic reporting

This is an important contribution and update to the knowledge of the Naupactini of the New World, specially for the naupactine fauna of the Andean Region and the South American transition zone.

The taxon sampling in the present contribution follows findings in del Río, Marvaldi & Lanteri (2012) and Lanteri & del Rio (2017). The genus-level relationships depicted here agree with del Río, Marvaldi & Lanteri (2012) but no with those of Lanteri & del Rio (2017). This is a important point of discussion that is missing in the paper (see comments on PDF).
The character statistics (l, ci, and ri) are not mentioned. This can be added in Table 1 (see PDF).
The figures are very helpful. I suggested some small changes in the figure captions (see PDF).

Experimental design

no comment

Validity of the findings

no comment

Additional comments

no comment

·

Basic reporting

The manuscript is clear and well written. It includes relevant references for most of the aspects discussed. My only complain (very minor) is that figure legends were not included in the main word document, but only visible in the final PDF compiled by the journal's platform.
Some of the figures might be improved by increasing the size of the images, reducing blank space around relevant parts.

Experimental design

The manuscript presents new informaion: a new species, lectotype designations, a new synonym, descriptions of males and immature stages.
The methods are clearly presented.

Validity of the findings

The manuscript presents relevant supporting information.

Additional comments

The manuscript is very clear and well supported. I only have a few suggestions marked throughout the text, especially concerning morphological terminology and consistency on the use of that terminology. I also added a few questions in the discussion that the authors might be able to tackle with the information presented. I also suggest to summarize information on ecology/associated plants for each of the treated species, since that information is presented already, but is sort of diluted in the material examined section, and might be relevant to discuss the implications of the newly found phylogenetic relationships. See comments in both the main word document and the PDF (only in the figure captions for this one).

---

## Round 0.2 · Minor Revisions

The manuscript represents an appropriate corrected version of the previously submitted. I attach herein a file with some minor formal corrections. Please check them an accept them if you agree with them.

---

## Round 0.3 · accepted · Accept

Your manuscript is OK, I am accepting it.